# Restoring oak forests through direct seeding or planting: Protocol for a continental-scale experiment

**Alexandro B. Leverkus**[1,2]*, **Laura Levy**[1], **Enrique Andivia**[3], **Peter Annighöfer**[4], **Bart De Cuyper**[5], **Vladan Ivetic**[6], **Dagnija Lazdina**[7], **Magnus Löf**[8], **Pedro Villar-Salvador**[9]

**1** Departamento de Ecología, Facultad de Ciencias, Universidad de Granada, Granada, Spain, **2** Laboratorio de Ecología, Instituto Interuniversitario de Investigación del Sistema Tierra en Andalucía (IISTA), Universidad de Granada, Granada, Spain, **3** Departamento de Biodiversidad, Ecología y Evolución, Universidad Complutense de Madrid, Madrid, Spain, **4** Professorship of Forest and Agroforest Systems, Technical University of Munich, Freising, Germany, **5** Research Institute for Nature and Forest, Brussel, Belgium, **6** Faculty of Forestry, University of Belgrade, Belgrade, Serbia, **7** Latvian State Forest Research Institute Silava, Salaspils, Latvia, **8** Southern Swedish Forest Research Center, Swedish University of Agricultural Sciences, Lomma, Sweden, **9** Grupo de Ecología y Restauración Forestal (FORECO), Departamento de Ciencias de la Vida, Universidad de Alcalá, Alcalá de Henares, Madrid, Spain

* leverkus@ugr.es

**Funding:** This collaborative experiment lacks specific funding and was designed to be as small, low-cost, and easy-to-implement as possible for each participant. The resources necessary to

## Abstract

The choice of revegetating via direct seeding or planting nursery-grown seedlings influences the potential stresses suffered by seedlings such as herbivory and drought. The outcome of the balance between both revegetation methods may ultimately depend on how species identity and traits such as seed and seedling size interact with environmental conditions. To test this, we will conduct a continental-scale experiment consisting of one mini-experiment replicated by multiple participants across Europe. Each participant will establish a site with seeded and planted individuals of one or more native, locally growing oak (*Quercus*) species; the selection of this genus aims to favour continental-scale participation and to allow testing the response of a widely distributed genus of broad ecological and economic relevance. At each site, participants will follow the present protocol for seed collection, seeding in the field, nursery cultivation, outplanting, protection against herbivores, site maintenance, and measurement of seedling performance and environmental variables. Each measurement on each species at each site will produce one effect size; the data will be analysed through mixed-effects meta-analysis. With this approach we will assess the main effect of revegetation method, species, plant functional traits, and the potential effect of site-specific effect moderators. Overall, we will provide a continental-scale estimate on the seeding *vs.* planting dilemma and analyse to what extent the differences in environmental conditions across sites, seed size, functional traits, and the phylogenetic relatedness of species can account for the differences in the effect of revegetation method on seedling performance across study sites and species.

establish and monitor each experimental site as described in this document will be made available by each participant. Funding for the coordination of this experiment is available from grant RTI2018-096187-J-100 from FEDER/ Spanish Ministry of Science, Innovation and Universities. The funders had no role in study design, data collection and analysis, decision to publish, or preparation of the manuscript.

**Competing interests:** The authors have declared that no competing interests exist.

## Introduction

Roughly 2 billion ha of land are in need of ecological restoration [1]. In recognition of the importance of restoration for climate mitigation and the provision of ecosystem services [2, 3], the UN Decade on Ecological Restoration provides an opportunity to advance the science and practice of restoration ecology [4].

Revegetation is at the core of restoration actions and it has been conducted for centuries (e.g., [5]), yet its success is not necessarily guaranteed. Revegetation failure often results from adverse biotic or abiotic conditions [6], both of which can be influenced by revegetation method. Active revegetation frequently relies on the planting of nursery-grown seedlings. The alternative, direct seeding, is often discarded due to presumably low seedling establishment [7] and the resulting loss of valuable seed material and waste of resources. Planting can speed seedling growth, it avoids seed predation and lack of seedling emergence, and it reduces the vulnerability of recently emerged seedlings to stress factors. Seeding, on the other hand, is easier and cheaper to carry out [6]. For some species, such as oaks (*Quercus* spp.), both methods are possible, yet considerable debate still surrounds the question of which method can maximise outcomes [6].

Root morphology is affected by the choice of revegetation method, with potential implications for the access to soil resources of plant species that develop deep roots. This is the case in oaks, as the tap root of nursery-grown seedlings is often pruned or deformed [6]. This may reduce access to soil resources, and ultimately hinder seedling performance under water shortage, with effects that can last until adulthood [8, 9]. The success of revegetation in terms of field performance of seedlings may thus depend on the interaction between species' traits such as root depth (which may in turn be related to seed size) and environmental conditions such as climate and soil characteristics. However, the preliminary outcomes of an ongoing systematic review [10] suggest that this question has not been empirically addressed to date.

Here we outline the protocol for an experiment designed to address the seeding *vs.* planting dilemma and identify the drivers of differences in effects at a continental scale. It will be conducted at multiple sites across Eurasia encompassing large climate differences. The experiment will aim at: i) providing continental-scale evidence on the balance between seeding and planting for oaks, ii) testing whether this balance depends on species choice, and iii) assessing whether climatic conditions and soil characteristics interact with species traits to explain heterogeneity across sites. The experiment, which will begin in Autumn 2021, has been discussed in the PEN-CAFoRR Cost action (http://www.pen-caforr.org/) and it has received widespread support, although participation is open also beyond the members of the action. Overall, the study shall produce new knowledge for improving forest and agroforestry ecosystem restoration.

## Materials and methods

### Overall description

**Target intervention and species.** We will test the effect of revegetation through direct seeding in the field *vs.* planting of seedlings previously grown in containers in the nursery. The experiment will encompass any native oak (*Quercus*) species with local populations. The selection of this widespread genus aims to promote broad participation and flexibility in the selection of species with local seed sources, testing the effect of revegetation method across oak phylogeny, and addressing the seeding *vs.* planting dilemma for an ecologically and economically relevant genus.

**Experimental design.** The experiment will consist of multiple sites (in the dozens) across Europe (yet open to potential sites across Eurasia; Fig 1). Sites will be analogous to a "study" in

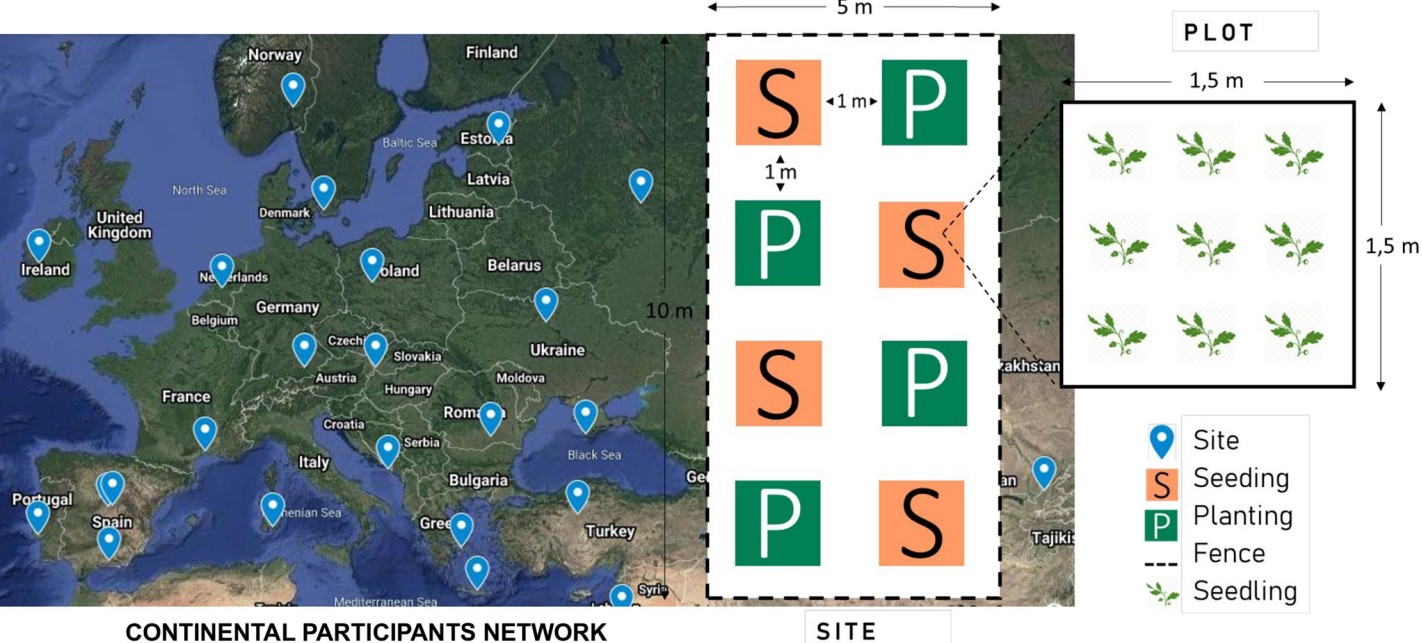

**Fig 1. Study design.** Across Eurasia, multiple sites will be established by voluntary participants (marks on the map). Each site will encompass at least eight plots, which constitute four replicates of two revegetation methods (S = seeding, P = planting). Each plot will contain nine plant-points with a target number of one sown or planted individual. A site may contain more species, in which case the number of plots would be 8 × the number of species. The map shows hypothetical locations; an updated map showing the location of sites can be found here.

meta-analysis, as each site will produce effect sizes [11]. In each site, we will grow individuals in the field of one or more oak species through both seeding and planting. There will be variability across sites in species, climate, and soil characteristics, which will allow assessing the interaction of species and environmental parameters with the target intervention (revegetation method). We aim at minimizing additional heterogeneity by strictly following this protocol across all sites.

The study has two key components: (1) the establishment and monitoring of sites by participants, including the commitment to produce certain deliverables, and (2) the coordination of the experiment and data analyses by the organizers (Fig 2).

## Procedure for participants at each site

Participants have so far registered their preliminary interest in participating in the experiment through an online survey. As of 23 September 2021, 58 respondents have filled this form and another 12 people have contacted us per email. The possibility for participation will be open as long as the phenology at new sites allows for seed collection and seeding and the prior posting of materials by the organisers (presumably until October–November 2021, depending on local conditions).

**Site design and treatments.** Each site will contain eight plots of 1.5 × 1.5 m per species (with a minimum of one species), each of which will contain nine plant-points with either sown or planted seedlings (8 plots x 9 plants = target number of 72 plants per species; Figs 1 and 3). These eight plots will comprise four replicates of each of the two revegetation methods. The seeding plots will be established in autumn or winter 2021 and the planting plots in early 2023 (details below). To obtain the target number of plants, a higher number (72) will be cultivated in the nursery and an excess of individuals (also 72) will be seeded in the field (two

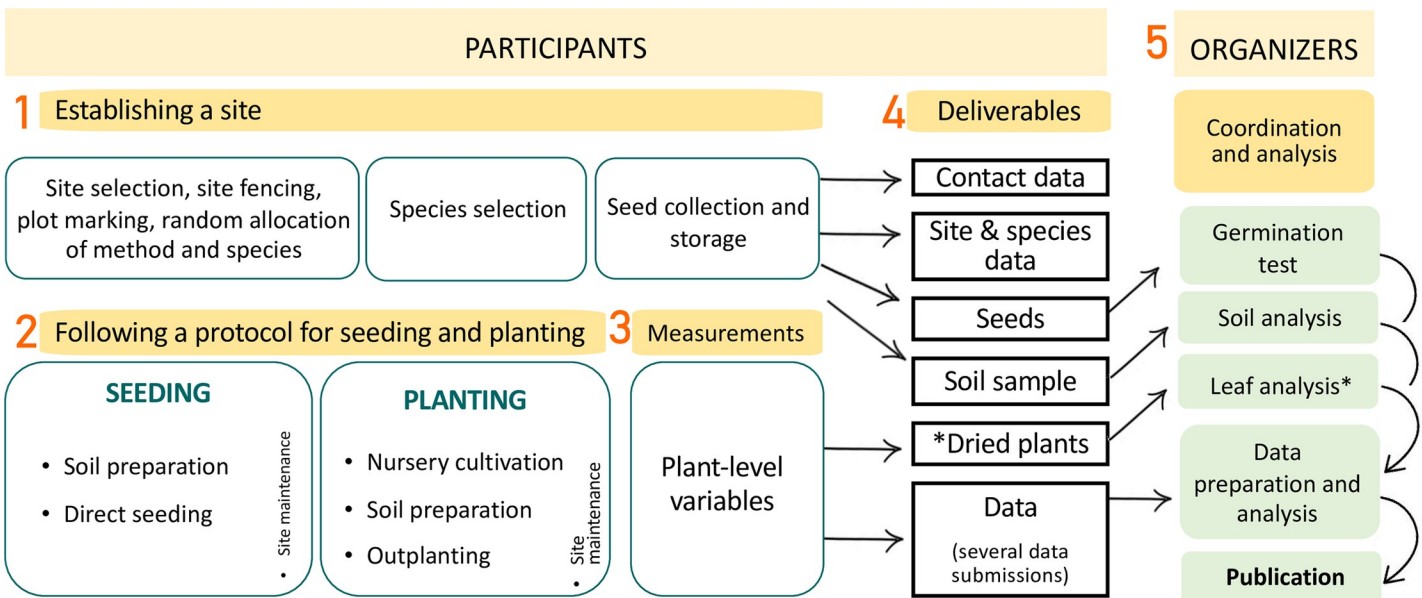

**Fig 2. Schematic representation of the experiment.** The activities are grouped by those to be conducted by each participant and those for the organizers. The deliverables in the rectangular boxes will be sent to, and processed by, the organizing team at the University of Granada. *Dried plants may be posted at the end of the experiment for chemical analysis in case of availability of additional funding.

acorns per seeding point). The total number of plots per site will thus result from multiplying eight plots by the number of species (see Fig 3 for an example with two species). Ideally, sites should include as many species as possible.

**Establishment of a site.** Each site will encompass a surface large enough to establish at least eight plots of 1.5 x 1.5 m. Plots may or may not be contiguous (see Fig 3) but they must be

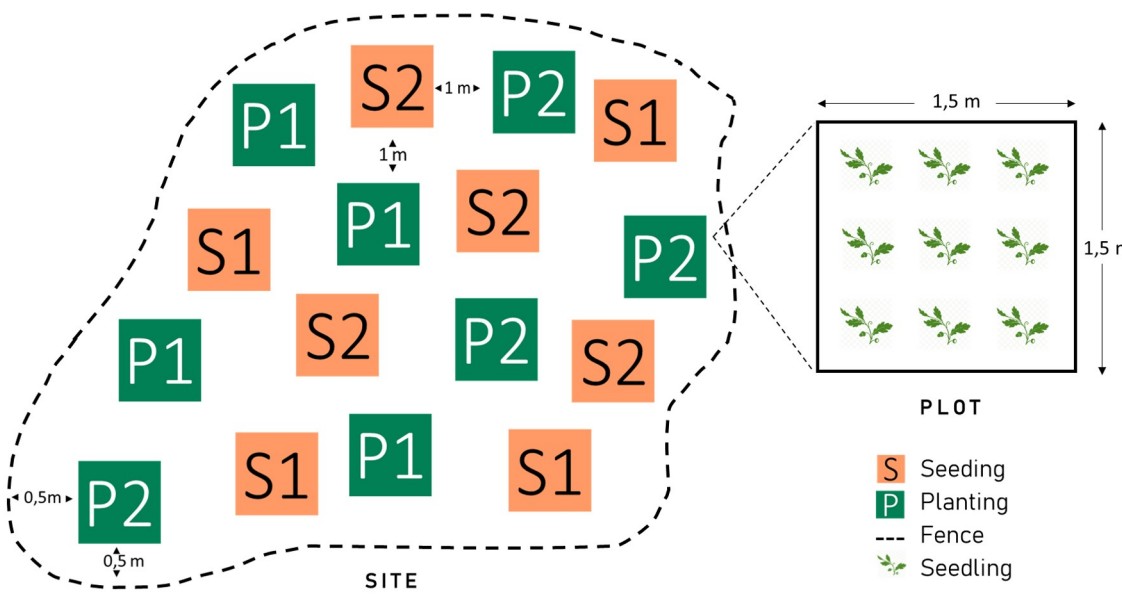

**Fig 3. Design of a site of irregular shape and with two species (indicated as 1 and 2).** The experimental combinations must be randomly allocated to all plots, thereby ensuring that plots of different species and method are spatially intermingled.

within approximately 100 m from each other, have similar conditions, and be at least 1 m apart from each other and 0.5 m from the edge of the site. For a site with one species, the minimum size for a rectangular-shaped site would thus be 5 × 10 m (Fig 1). The allocation of revegetation method and species to each plot will be made randomly at each site. Each site must meet the following conditions:

- Sites must be located in an open area. Abandoned fields, clearcuts, burnt areas, cleared windthrows, and other open areas are allowed. However, highly degraded areas or sites with heavily altered soil, such as roadsides, mines, and polluted sites, will not be considered. The conditions of the site (soil type, bedrock, elevation, aspect, etc.) should be representative of the area (e.g. avoid selecting a patch of rare bedrock or the top of the highest mountain of the region).

- The site should be as flat as possible, with a maximum slope of ca. 10%. Flat terraces on an otherwise steeper mountain side are allowed.

- There should be no trees or shrubs inside the plots.

- Each plot must be clearly identified and marked and will later randomly be allocated one of the combinations of revegetation method and species (in case more than one species is used).

- Plants must be protected from herbivores, preferably through fences around the whole site, but individual protection may also be used. Insecticide may be used if necessary; the product to be used should be agreed on with the organizers to avoid secondary effects. The experiment will strictly run under conditions of no browsing of shoot tissues. It is the responsibility of each participant to define how to achieve this.

- Note that each participant is responsible for the obtention of the necessary permits to establish their site.

## Deliverable 1: Communication of the location of experimental sites and species choice (October 2021)

- Willingness to participate should be communicated to the project coordinators as early as possible.

- The characteristics of sites, species choice, and participants' details must be registered through a spreadsheet sent by the coordinators to those who have indicated their interest in participating. New sites should ideally be located as far away as possible from other already recorded sites, or cover a different set of environmental conditions. The objective is to maximize the geographic distribution of sites and avoid spatial/ environmental clusters.

- Along with the site coordinates, the number and identity of the local oak species expected to be used should be communicated, along with a list of potential alternative species. These alternative species would be used in case of insufficient seed production, in case some species would be used in fewer than 3 sites, or under other contingencies.

**Procedure for acorn collection, selection, and storage (Autumn 2021).** The seeds of the local target species will be collected from local populations growing at an elevation, aspect, and substrate as similar as possible to that of the experimental site. Upon identification of a source population, mature and healthy-looking acorns will be collected on a minimum of 10 parent

trees, evenly distributed across parents (similar number of seeds from each tree). If necessary, local seeds may exceptionally be bought. To account for seed selection, the need for a germination test (see below), and seed/ seedling losses, seeds will be collected in excess, with a target of at least 500 acorns per site and species. Participants may decide to collect more in case a visual inspection suggests that a high proportion of acorns is affected by insects.

After collection, acorns will be selected with the flotation method [12], which consists of introducing them in water and eliminating those that float. A second selection of all acorns through flotation may be conducted before the time when acorns are first used for either a) seeding, b) cultivating in the nursery, or c) conducting the germination test; the resulting acorn lot shall be considered definitive. As soon as a subset of acorns is used for one of these three processes, no further non-random acorn selection shall be conducted.

The selected acorns will be stored in zip polyethylene plastic bags with a thickness of no more than 50 μm (which will be posted by the organizers to all participants along with the seed protectors), and these will be placed in a refrigerator between 1 and 4˚C. Before storage, the surface of the acorns must be dried by leaving the acorns to dry out for 24h at laboratory temperature. Subsequent handling of acorns will be done as swiftly as possible to minimize the risk of seed desiccation or fungal attack during storage. Acorns should be inspected periodically during storage to detect possible rottenness. Bags should not be stacked while stored to facilitate gas exchange, and storage boxes should be perforated to facilitate ventilation. If acorns germinate during storage, they may still be used as long as the radicle is not damaged or longer than 2 cm.

**Deliverable 2: Seeds and soil samples (deadline: November–December 2021).** The participants will post one parcel per site to the organizing team, containing:

- 150 acorns per species. They will be placed inside the afore-mentioned polyethylene zip bags and cushioned to avoid physical damage.

- A composite soil sample from the experimental site. To produce the sample, three holes of 20 cm depth will be dug at different points within the site. The soil from the three samples will be thoroughly mixed, and 1 kg of the mixture will be placed in a bag such as those used for acorn storage. If the sample contains small stones, 1.5–2 Kg should be taken, as these should be removed in the laboratory prior to posting the sample. The sample will be allowed to dry on laboratory paper under indoor conditions for 1 week previous to posting it. As a reference, photographies of the soil should be taken in the field (to show the soil under natural conditions) and as soon as the sample is left to dry in the laboratory (to show the stoniness of the sample before stones are removed and the physical structure of aggregates before they dry).

The parcel will be sent as soon as possible after the obtention of the acorns and the soil sample through an express courier service to avoid the deterioration of the soil or the loss of viability of the acorns during transportation. Participants are kindly asked to cover the cost of the courier service, yet in case this was impossible, please contact the organizing team. The address is:

<div align="center">

Alexandro B Leverkus
Departamento de Ecología, Facultad de Ciencias
Campus Fuentenueva s/n
18071 Granada, Spain

</div>

**Soil preparation (Autumn 2021–winter 2022 for seeding; winter-spring 2023 for planting).** Soil preparation for both revegetation methods will consist of the clearing of vegetation inside the plots and the excavation of a $40 \times 40 \times 40$ cm hole for each seedling or pair of seeds.

This can be achieved either manually or mechanically as long as these dimensions are reached and not exceeded. The holes for seeding will be excavated in the first year (i.e., right before seeding) and those for the planting treatment in the following year (i.e., right before planting). In case of mechanical soil preparation, the spatial arrangement of the experiment must allow the machinery to access the planting plots in the second year without disturbing the plots that were seeded in the first year. Each plant-point will be marked to identify individual plants.

**Procedure for direct seeding (Autumn 2021 or winter 2021–22).** Before seeding, the acorns will be soaked in water for 24 h. The previously excavated soil will be placed back in the hole, and seeds will be placed at ~3–4 cm depth within an acorn protector (see below for details). To secure a sufficient number of seedlings, two acorns will be placed next to each other in each seed-point, each with an individual protector. The date of seeding will be selected based on local best practice but, among the range of possible dates, the earliest will be pre-ferred. Either on the same or on the following day, 2L of water will be applied to each seed-point.

Seeds will be placed in the ground inside of seed protectors to avoid their predation by small animals. We will use commercial seed protectors (*seed shelters* [13, 14]), which will be posted by the organizers to all participating sites. The protectors consist of two truncated pyra-mids joined at their larger opening; the stem and root can exit through the small openings at the top and bottom but the dimensions of these holes preclude access to rodents (Fig 4). The two halves are completely filled with local soil and one acorn in the middle prior assembly of the two units, taking care that they are completely full of soil. The complete device is then placed in the ground, with the upper opening 1 cm beneath the surface. The material currently under commercialization is not biodegradable, so the seed shelters may need to be removed at the end of the experiment.

**Procedure for nursery cultivation (Winter 2022- winter/spring 2023).** At a date close to that of seeding in the field (+/- 2 weeks), nursery cultivation will be initiated. Cultivation may be conducted in a nursery managed by participants' research institution or at any other nurs-ery that is publicly or privately owned. The number of seedlings to be cultivated will be the same as the number of acorns to be seeded in the field, which is twice the target number of plants (72 individuals per species and method).

The seedlings will be grown in common plastic containers of 300 (250–300) mL and ~15 cm depth, at a density of $<$350 plants m$^{-2}$. The substrate to be used will consist of a mixture of

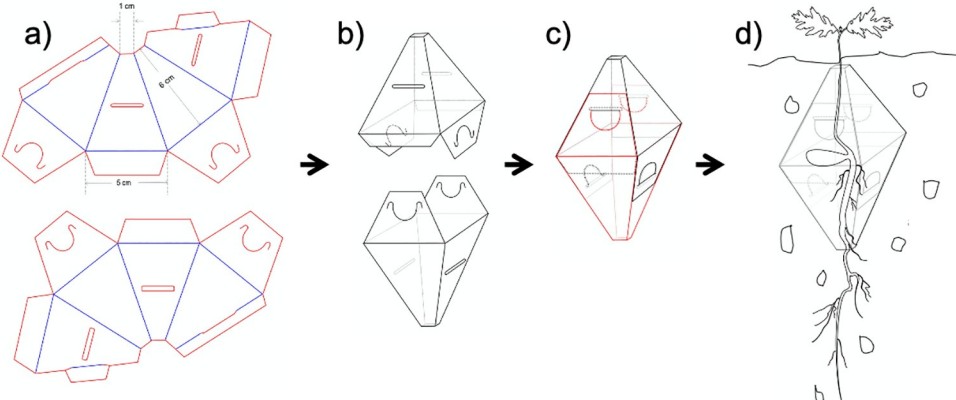

**Fig 4. Seed shelters to be used to protect acorns from small vertebrate predators.** Before assembling the two units, they are filled with local soil and an acorn is placed in the middle. Reproduced from [12] with permission from Springer-Nature.

blonde peat and vermiculite at a proportion of 3:1. Prior to this process, acorns will be soaked in water for 24 h. Once a container is filled with substrate, one acorn will be placed as horizontally as possible on the surface and gently pushed down to a depth of approximately 1 cm. It will then be covered and the surface above will be gently compacted. Watering will be applied on the same day.

The seedlings will be grown outdoors under full sunlight for as long as possible (taking care to protect the containers from potential acorn predators such as rodents and birds). However, seedlings should be placed in a greenhouse for the necessary time in winter. Fertilization will be done by mixing slow-release fertilizers. As a baseline, the recommendation is a fertilizer of 12–14 month duration; N:P:K 18-6-8 plus micronutrients (for instance Nutricote Total®, Arista LifeScience) at 7 g L$^{-1}$ applied at the beginning of the cultivation period in the nursery. In case of poor seedling growth or symptoms of nutrient deficiency, the plants must be manually supplied with further slow-release fertilizer. The rate and the need for additional fertilization may need to be adapted to the needs of local species and climate; it is strongly recommended to contact professionals in the sector such as nurseries for advice, or ask a nursery to produce the seedlings. Watering will be applied on demand to ensure an optimal growth.

**Procedure for outplanting (Winter/ spring '23).**    At one year of age, containerized seedlings will be planted in the field and placed in the previously excavated holes (see *Seedling selection*, below). This will be done at the time of the year that is optimal given local best practice but preferentially in late winter or early spring; hole digging should occur shortly before. Preferentially, seedlings grown close to each other in the nursery will be placed in different field plots (to avoid seedlings that had similar light or water conditions in the nursery being placed together in a field plot). The excavated soil will be used to re-fill the part of the hole remaining empty and gently pushed down to avoid remaining holes. Either on the same or on the following day, 2L of water will be applied to each plant. The person(s) conducting the seeding should be the same doing the outplanting to ensure similarity in the methods.

**Seedling selection.**    As indicated before, both the seeding and the planting treatments will start with twice the target number of individuals to ensure that sufficient plants are present for the actual experiment. As a consequence, half of the individuals will need to be selected to be part of the experiment. This selection will be done at the time of the outplanting of nursery seedlings under the following procedure:

- Seeding treatment: In those plant-points in which the two seeded acorns produced an emerged seedling, the tallest will be kept and the shortest will be carefully removed.

- Planting treatment: Among pairs of seedlings growing next to each other in the nursery, the tallest will be used for outplanting and the shortest will be kept in the container.

The size of all selected seedlings–both seeded and planted individuals–will be measured at the time of transplanting (see *Measurements*, below). Additionally, among the individuals that are not used in the experiment, the shoot tissues of 15 seeded individuals and 15 planted individuals will be oven-dried and weighed. One month after planting, mortality due to transplant shock will be assessed, and dead nursery-grown seedlings will be replaced with "extra" seedlings from the nursery, following the same procedure. The number of replaced seedlings will be noted and they will be measured.

**Site maintenance.**    Herbs growing inside the plots need to be removed manually on a periodical basis. The frequency and timing of weeding must be decided according to the cover of herbs at each site, as follows. Weeding should occur on the same date for the whole site, no later than the time when herbs have grown to cover ~80% of the surface of the plots. This

maintenance should begin in early 2022, when the seedlings are emerging from seeded acorns, and it must be conducted carefully to avoid damaging the oaks that are the target of the experiment. Additionally, in case needed, insecticides may be used for application on the ground against ants or on shoot tissue against other insects; prior to application, the identity of the product and the reason to use it must be agreed on with the organizers to avoid side-effects. Watering will not be applied except at the time of seeding or planting (see above).

**Deliverable 3: Data (various timings).** Participants are required to submit data measured at the individual plant level. For this, a spreadsheet will be prepared by the organizing team and sent to the participants at each measurement time. In the spreadsheet, each row will constitute one individual plant, which in every case will be identified through four columns: site name, treatment (seeded or planted), plot number (1–4), and plant-point number (1–9). The remaining columns will be the measured responses, which may include the following at the given points in time:

1. After the emergence of seeded individuals, in mid-2022. Number of seedlings emerged at each plant-point (0, 1 or 2).

2. Time of outplanting of nursery seedlings, in early 2023. For seeding plots: number of live seedlings remaining at each plant-point (0, 1 or 2). Among the rest of the cultivated seedlings, wet and dry root and shoot mass will be measured on a random subset of 15 planted and 15 seeded individuals. For this, the indicated seedlings will be cut at the base, their roots and shoots separately weighed, then oven-dried (minimum 3 days at 60˚C), and weighed again. The rest of data will be measured on all seedlings that remain after the seedling selection procedure described above: stem height (from the ground level to the stem apex), and stem diameter (in two perpendicular measurements), which should be measured at the same point during each of the measurements. In case more than one stem emerged from the seed shelter, the height and diameter of all living stems will be measured. The same measurements will be made for replacement seedlings at the time of replacement (see the end of the section *seedling selection*, above).

3. After the first growing season. After summer 2023, mortality will be assessed, as well as stem height and diameter (as described above).

4. After the first winter. After winter 2023–4, mortality will be assessed.

5. After the second growing season. After summer 2024, mortality will be assessed, as well as stem height and diameter. Shoot wet and dry mass will be measured on either all individuals or a subset (to be decided depending on the proportion of survivors). All measurements will be conducted in the way described in point 2 above; wet mass must be measured in the field quickly after harvest.

6. More measurements on plant performance, other ecophysiological variables, or environmental covariates are possible (including chemical analyses of the harvested seedlings) if funding is made available or if some person from the network takes the initiative.

**Key material requirements for each participant.** The key material needs identified required for each participating site include the following:

- Local source of acorns of all target species

- Site for planting; it must be accessible, protected against herbivores, and secure

- Greenhouse or a nursery facility for plant cultivation

- Transport to the field including for containerized seedlings

- Materials for digging, marking plots and tagging plants

- Oven

- Scales: laboratory and field

- Ruler and calliper

- Computer, printer, writing materials

- Desired: funding for courier service

### Procedure for the organizers

**Dissemination.**   The experiment was initially proposed to be conducted as part of the PEN-CAFoRR COST action, of which all the authors of this protocol are members and/or leaders of the action or of particular work packages. Following a positive initial response to the idea, more specific information has been circulated among all action members, and a survey was sent for participants to indicate willingness to participate and the geographic location of potential sites. The survey was circulated among participants' colleagues until ca. 70 participants have registered. Several brochures with information have been sent so far (see S1–S4 Figs). For visual identity, a logo has been created (**Fig 5**).

Participant recruitment is still open at the time of submission of this protocol and as long as there is sufficient time to organize the creation of a site under optimal conditions. Meetings will be held to solve final questions in early autumn, by when this protocol will be provided to all participants. A website is being created, which will contain relevant updates, and it will be hosted at the website of the Department of Ecology, University of Granada.

**Sending and receiving materials.**   In early autumn 2021, the organizing team will post parcels with seed shelters for the experiment and plastic bags for acorn storage and transportation to all participating sites. In late autumn, parcels will be received from participants with soil samples and seeds for further handling.

**Germination test (Autumn 2021 –spring 2022).**   From each seed lot, 150 acorns will be sent per mail to the study coordination site at the University of Granada. Upon reception, they will be stored under refrigeration. At the end of November 2021, a germination test will be initiated with 100 individuals per species and site, with the aim of being able to differentiate emergence percentage in the field from seedlot potential. The test will be conducted following the recommendations of the International Seed Testing Association. All acorns will be placed under optimal conditions for germination and the percentage of non-germinated acorns will be assessed after 4 weeks. Prior to the test, all acorns will be soaked in water for 24 h. The remaining 50 acorns per species and site will be oven-dried and their mass will be measured.

**Soil analyses.**   One composite soil sample will be sent to Granada from each site and be processed to assess basic soil properties including granulometry, cation exchange capacity, water holding capacity, and the content of key nutrients such as P and N as well as C. These analyses will prospectively be conducted at the laboratory for soil science of the University of Granada. The data will be used to explore the extent to which the response of seedlings to revegetation method can be explained by differences in soil type.

**Statistical analyses.**   The data will constitute a multi-site study suitable for meta-analysis [15]. Each measurement on each species at each site will produce one effect size. The significance of an overall effect size of revegetation method on various measures of seedling performance will be tested. In case of significant heterogeneity in effect sizes among studies, which is

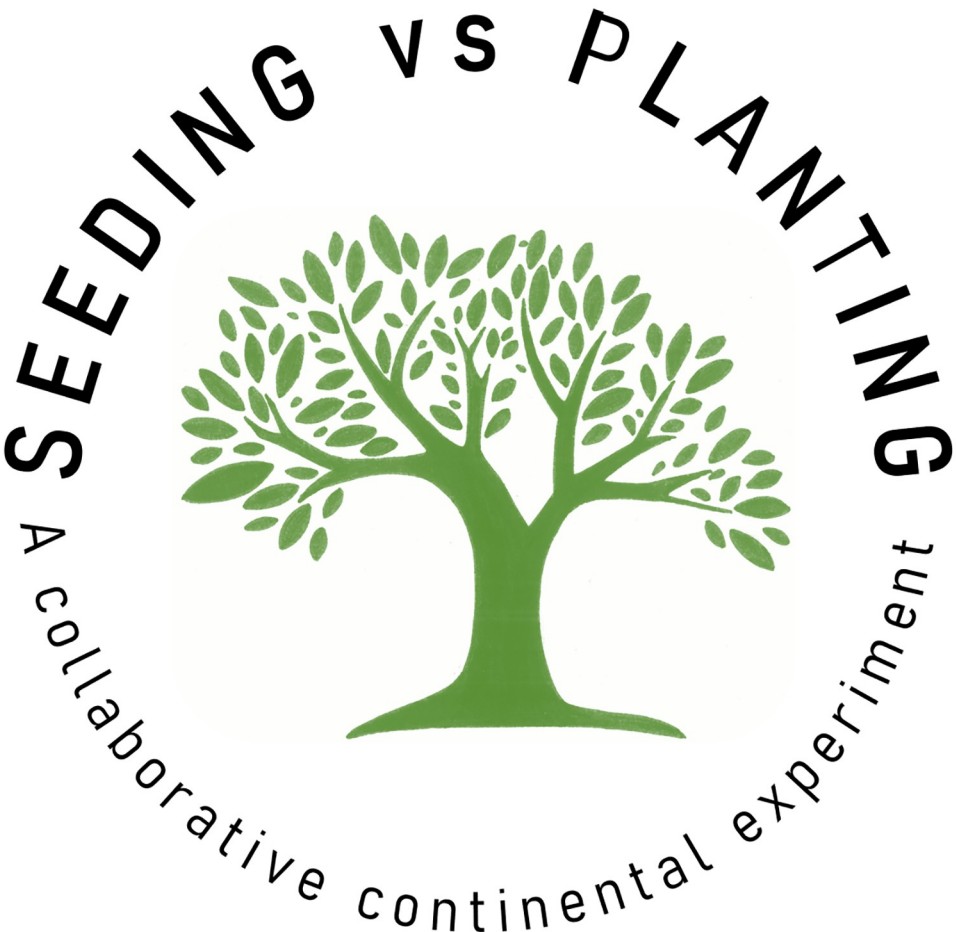

**Fig 5. Logo for the experiment.**

expected, the effect of covariates (variables that differ between data points, including soil parameters, mean site-level temperature to be obtained from www.worldclim.com, and species) will be modelled with mixed-effects meta-regression. Species' traits will also be explored as covariates, including mean seed mass measured from each lot and potentially other species-level traits obtained from databases such as BIEN or LEDA. The model will include site as a random effect, as well as the phylogenetic relatedness among species [e.g., [16]], which will be built with the V.PhyloMaker R package [17].

The complete timeline for the experiment is indicated in Fig 6.

## Discussion

### Control of bias

One key aspect of comparing direct seeding and seedling planting as revegetation methods is to avoid bias (for a discussion, see [10]). In many studies, bias results from comparing plots containing outplanted nursery seedlings with plots where direct seeding was conducted at the same time, and therefore the plants are younger in the latter plots, and seeds do not come from the same batch. In our experiment, the seedlings under both revegetation methods at each site will be produced from the same seed batches and grown at the same time. Seeds will be seeded

| Who | Action | 2021 | | 2022 | | | | 2023 | | | | 2024 | | | |
|---|---|---|---|---|---|---|---|---|---|---|---|---|---|---|---|
| | | summer | autumn | winter | spring | summer | autumn | winter | spring | summer | autumn | winter | spring | summer | autumn |
| All participants | Site preparation | | Soil preparation. Marking plots and plant-points | Weeding* | Weeding* | Weeding* | | Weeding* | Weeding* | Weeding* | | Weeding* | Weeding* | Weeding* | |
| | Seed preparation | | Seed collection, selection, storage | | | | | | | | | | | | |
| | Seeding | | Seeding in the field | | Measure emergence | | | Initial seedling measurements | | Seedling measurements after growing season | | Assess survival after winter | | Seedling measurements after growing season | |
| | Planting | | Start nursery cultivation | | | | | Outplanting nursery seedlings & Initial seedling measurements. Dead-seedling replacement & measurement. | | Seedling measurements after growing season | | Assess survival after winter | | Seedling measurements after growing season | |
| | Deliverables | | Send site and species info. Post seeds and soils | | Data #1 | | | Data #2 | | Data #3 | | Data #4 | | Data #5 | Post dry plants (to be decided) |
| Coordinators | Germination test and soil analysis | | Start of germination test | Soil analysis | End of germination test | | | | | | | | | | |
| | Data management | Attract and inform participants | Disseminate full study protocol | | | Prepare emergence, soil and germination data | | | Prepare data | | Prepare data | | | Prepare and analyse data | |
| | Write-up | | | Write up introduction & methods | | | | Update introduction & methods | | | | | | Prepare results & discussion | |

**Fig 6. Timeline for the experiment.** The exact timing of seed collection, seeding, outplanting, weeding, and measurements will need adjustment to local phenology and must be decided by participants. *Weeding will be done when >80% of the plots are covered by herbs.

in the field in the same year as in the nursery, and the nursery-grown seedlings will later be outplanted to the field, thereby producing similarly-aged "siblings" grown under the two different methods. This approach may introduce a different source of bias, namely that of the conditions of the year of outplanting differing across the two revegetation methods. This would be problematic for studies based on a single study site, yet as the weather-at-outplanting effect is expected to be random across sites, there is no expectation that this would introduce systematic bias in the data and it can thus properly be dealt with using site-specific random effects in statistical analyses [11, 18], as we plan to do.

## Potential for additional treatments or measurements

The conditions outlined in this document are a minimum that is required for each site and they are sufficient for participation in the experiment. Additional treatments, measurements, etc., as well as the possible continuation of the experiment beyond the indicated timeframes, may be proposed by participants to the project coordinators. Upon agreement, proposals will be passed to the participants so that each can decide on whether to implement the new protocols.

## Publications and authorship

The first publication in a scientific journal resulting from the data of this experiment will be led by the coordinators of this initiative. The data will be made publicly available in an open

access repository along with the first publication, so anyone will be able to access the dataset and make additional use of it. Requests for additional use of the data prior to the first publication would be assessed by the coordinators. In general, single authors can consider using the data from individual sites, yet not simultaneously using data from more than one site for publications outside of the project framework. This has the objective of maximizing both the scientific outcome of the experiment as a whole and enhancing the academic output for all participants.

All persons who set up a site according to the protocol and produce all deliverables in proper form and time will be invited to co-author the resulting publication(s). The author list will consist of one person per species and site (strictly not more), with the possible addition of people who provide considerable assistance in other aspects of the project (for instance in soil analyses, germination test, data handling, considerable intellectual input, etc.). For co-authorship, it will be further required that every author at least reads the manuscript and explicitly approves its submission.

Participation in the study implies the acceptance of all the conditions established in this protocol.

## Supporting information

**S1 Fig. First brochure, sent to all participants of the PEN-CAFoRR COST action on 1 June 2021 to present the idea of the experiment.**
(PDF)

**S2 Fig. Second brochure, sent on 24 June 2021 to all people who registered their interest after sending the first brochure on 1 June 2021 to explain the design of a site.**
(PDF)

**S3 Fig. Third brochure, sent on 24 June 2021 to all people who registered their interest by this date to explain the timeline and requirements for participants.**
(PDF)

**S4 Fig. Fourth brochure, sent on 11 August 2021 to all people who registered their interest by this date to give guidelines on seed collection and storage.**
(PDF)

## Acknowledgments

We are grateful to the participants of the PEN-CAFoRR COST action, who have so far warmly welcomed this experiment by signing up to establishing local sites, and we look forward to fulfilling this collaboration. We thank A. Palma, J. Oliet and T. Morán for suggestions on improving this protocol.

## Author Contributions

**Conceptualization:** Alexandro B. Leverkus, Enrique Andivia, Magnus Löf.

**Funding acquisition:** Alexandro B. Leverkus.

**Methodology:** Alexandro B. Leverkus, Enrique Andivia, Peter Annighöfer, Bart De Cuyper, Dagnija Lazdina, Magnus Löf, Pedro Villar-Salvador.

**Project administration:** Alexandro B. Leverkus, Laura Levy.

**Visualization:** Laura Levy.

**Writing – original draft:** Alexandro B. Leverkus.

**Writing – review & editing:** Laura Levy, Enrique Andivia, Peter Annighöfer, Bart De Cuyper, Vladan Ivetic, Dagnija Lazdina, Magnus Löf, Pedro Villar-Salvador.

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
