## [Decision Letter · Decision Letter 0]

21 Sep 2021

PONE-D-21-24564Restoring vegetation through direct seeding or planting: Protocol for a continental-scale experimentPLOS ONE

Dear Dr. Leverkus, I have now received the comments from both reviewers. All of us are very excited about this large-scale experiment that aims to compare the effectiveness of seed sowing vs planting seedlings in revegetation plans. Both reviewers agree that the study is timely and the protocol is well-described. Also they provide some interesting advices that you may want to incorporate. Reviewer 1 provides some advices about the use of insecticides and limiting the experiment to a lower number of species. I guess, you have decided to not constrain the number of species in order to take into account the natural variabililty found in Eurasia. This would  allow to provide general guidelines. However, in the analyses section it could be highlighted that the meta-analyses will take into account species-specific effects (probably as a random factor). Now it is stated along other potential covariates in L446.  Reviewer 2 is concerned about sowing seeds and planting seedlings in different years. I understand your reasoning (this way one-year-old seedlings are grown in the same environmental conditions), but also that this could make comparisons difficult if environmental conditions strongly vary between years. Could you provide a little bit more details about this issue?Finally, since your current deadline is October i suggest to move it some weeks (if possible) so that the paper is published before. I will be willing to accept it as soon as possible. Kind regards, 

Teresa Morán

We look forward to receiving your revised manuscript.

Kind regards,

Teresa Morán-López

Academic Editor

PLOS ONE

“This collaborative experiment lacks specific funding and was designed to be as small, low-cost, and easy-to-implement as possible for each participant. The resources necessary to establish and monitor each experimental site as described in this document will be made available by each participant. Funding for the coordination of this experiment is available from grant RTI2018-096187-J-100 from FEDER/ Spanish Ministry of Science, Innovation and Universities”

Upon requirement from the editors after submission, I attach a confirmation letter on the indicated grant. Note that the grant is for a project on seeding vs planting of oaks but that the study described in this protocol was not originally included in the project.”

Reviewers' comments:

Reviewer's Responses to Questions

**Comments to the Author**

1. Does the manuscript provide a valid rationale for the proposed study, with clearly identified and justified research questions?

Reviewer #1: Yes

Reviewer #2: Yes

2. Is the protocol technically sound and planned in a manner that will lead to a meaningful outcome and allow testing the stated hypotheses?

Reviewer #1: Yes

Reviewer #2: Yes

3. Is the methodology feasible and described in sufficient detail to allow the work to be replicable?

Reviewer #1: Yes

Reviewer #2: Yes

4. Have the authors described where all data underlying the findings will be made available when the study is complete?

Reviewer #1: Yes

Reviewer #2: Yes

5. Is the manuscript presented in an intelligible fashion and written in standard English?

Reviewer #1: Yes

Reviewer #2: Yes

6. Review Comments to the Author

You may also provide optional suggestions and comments to authors that they might find helpful in planning their study.

Reviewer #1: Dear Alexandro,

Your manuscript "Restoring vegetation through direct seeding or planting: Protocol for a continental-scale experiment" provides a clear and detailed methodology to test how seeds and seedlings interact with a range of environmental conditions across Eurasia.

The protocol is clear and well-written. You and your co-authors provide all the information needed to replicate the experiment across multiple sites and provide an overview of the data analysis that will be used.

I find the protocol very useful and I am looking forward to see your results to understand better how trees species and environmental parameters interact.

I only have two comments regarding the protocol and I think if you specify these things further, your experiment and data will be more robust. Please see comments below:

1. I think it would be better if you select a couple of insecticides that will be allowed to be used for the experiment. There are lots of products out there and if each participant chooses what they think is best, you might be adding another source of variation. I would suggest you and your co-authors choose one or two insecticides that could be applied if needed.

2. Thinking about reducing variation, I would also suggest you choose one or two species of Quercus to be used across the study sites. We know we can find different physiological responses from different individuals of the same species, so opening the variation to a whole genus could make the drawing of conclusions harder. What about using Quercus robur and/or Q. petraea? If you choose the same species, you could also collect very interesting data about the different responses of the same species to the range of environmental conditions present across your study sites.

Overall I enjoyed reading your protocol and I think is a great idea to test on a continental scale.

Reviewer #2: PLEASE SEE ATTACHED DOCUMENT FOR SPECIFIC COMMENTS

The paper is well written and clear, and the topic is very interesting. I’m sure that the outputs of this multi experiment will contribute effectively to optimize the results of forest planting. Congratulations to the authors.

But I must focus the attention on the question of the bias control. Authors propose that sowing must be conducted one year before than planting while seedlings of the same seedlot grow in the nursey. This avoid seeds coming from different genetics. However, this mean that sowing and planting will be conducted in different years. Authors argue that this schedule provides comparison performance of the same age seedlings. However, planting and sowing at different years can lead to huge inter-annual differences in environmental conditions. If the study aims at linking environmental conditions of the planting site with size effect of planting versus direct seeding, then comparing performance in two different years is not very appropriate. In many environments, inter-annual differences can be larger than inter-sites one. Besides, the mentioned “siblings” are quite different in terms of root architecture, nutrients, SLA, dry mass, carbohydrate reserves and so on. Those differences plus environmental ones from one year to another could be higher than argued differences of age. Optimal solution is using the same seed lot but storing the acorns for one year to sow them in the second year along with planting. See for instance: Oliet, J.A.; Vázquez de Castro, A.; Puértolas, J. 2015 Establishing Quercus ilex under Mediterranean dry conditions: sowing recalcitrant acorns versus planting seedlings at different depths and tube shelter light transmissions New Forests 46: 869-883

7. PLOS authors have the option to publish the peer review history of their article (what does this mean?). If published, this will include your full peer review and any attached files.

Reviewer #1: **Yes: **Ana C Palma

Reviewer #2: **Yes: **Juan A. Oliet

---

## [Author Response · Author response to Decision Letter 0]

6 Oct 2021

Please find our response to reviewer and editor comments as an attached .docx file.

---

## [Editor Report · Decision Letter 1]

21 Oct 2021

Restoring oak forests through direct seeding or planting: Protocol for a continental-scale experiment

PONE-D-21-24564R1

Dear Dr. Leverkus,

We’re pleased to inform you that your manuscript has been judged scientifically suitable for publication and will be formally accepted for publication once it meets all outstanding technical requirements.

Kind regards,

Teresa Morán-López

Academic Editor

PLOS ONE

Additional Editor Comments (optional):

Congratulations, for such a nice protocol work. I hope we are still on time!

Best, 

Tere

---

## [Editor Report · Acceptance letter]

26 Oct 2021

PONE-D-21-24564R1 

Restoring oak forests through direct seeding or planting: Protocol for a continental-scale experiment 

Dear Dr. Leverkus:

I'm pleased to inform you that your manuscript has been deemed suitable for publication in PLOS ONE. Congratulations! Your manuscript is now with our production department. 

Kind regards, 

on behalf of

Dr. Teresa Morán-López 

Academic Editor

PLOS ONE